# Discovering Structure in High-Dimensional Data Through Correlation Explanation

**Greg Ver Steeg**
Information Sciences Institute
University of Southern California
Marina del Rey, CA 90292
gregv@isi.edu

**Aram Galstyan**
Information Sciences Institute
University of Southern California
Marina del Rey, CA 90292
galstyan@isi.edu

## Abstract

We introduce a method to learn a hierarchy of successively more abstract representations of complex data based on optimizing an information-theoretic objective. Intuitively, the optimization searches for a set of latent factors that best explain the correlations in the data as measured by multivariate mutual information. The method is unsupervised, requires no model assumptions, and scales linearly with the number of variables which makes it an attractive approach for very high dimensional systems. We demonstrate that Correlation Explanation (CorEx) automatically discovers meaningful structure for data from diverse sources including personality tests, DNA, and human language.

## 1   Introduction

Without any prior knowledge, what can be automatically learned from high-dimensional data? If the variables are uncorrelated then the system is not really high-dimensional but should be viewed as a collection of unrelated univariate systems. If correlations exist, however, then some common cause or causes must be responsible for generating them. Without assuming any particular model for these hidden common causes, is it still possible to reconstruct them? We propose an information-theoretic principle, which we refer to as "correlation explanation", that codifies this problem in a model-free, mathematically principled way. Essentially, we are searching for latent factors so that, conditioned on these factors, the correlations in the data are minimized (as measured by multivariate mutual information). In other words, we look for the simplest explanation that accounts for the most correlations in the data. As a bonus, building on this information-based foundation leads naturally to an innovative paradigm for learning hierarchical representations that is more tractable than Bayesian structure learning and provides richer insights than neural network inspired approaches [1].

After introducing the principle of "Correlation Explanation" (CorEx) in Sec. 2, we show that it can be efficiently implemented in Sec. 3. To demonstrate the power of this approach, we begin Sec. 4 with a simple synthetic example and show that standard learning techniques all fail to detect high-dimensional structure while CorEx succeeds. In Sec. 4.2.1, we show that CorEx perfectly reverse engineers the "big five" personality types from survey data while other approaches fail to do so. In Sec. 4.2.2, CorEx automatically discovers in DNA nearly perfect predictors of independent signals relating to gender, geography, and ethnicity. In Sec. 4.2.3, we apply CorEx to text and recover both stylistic features and hierarchical topic representations. After briefly considering intriguing theoretical connections in Sec. 5, we conclude with future directions in Sec. 6.

## 2   Correlation Explanation

Using standard notation [2], capital $X$ denotes a discrete random variable whose instances are written in lowercase. A probability distribution over a random variable $X$, $p_X(X = x)$, is shortened

to $p(x)$ unless ambiguity arises. The cardinality of the set of values that a random variable can take will always be finite and denoted by $|X|$. If we have $n$ random variables, then $G$ is a subset of indices $G \subseteq \mathbb{N}_n = \{1, \ldots, n\}$ and $X_G$ is the corresponding subset of the random variables ($X_{\mathbb{N}_n}$ is shortened to $X$). Entropy is defined in the usual way as $H(X) \equiv \mathbb{E}_X[-\log p(x)]$. Higher-order entropies can be constructed in various ways from this standard definition. For instance, the mutual information between two random variables, $X_1$ and $X_2$ can be written $I(X_1 : X_2) = H(X_1) + H(X_2) - H(X_1, X_2)$.

The following measure of mutual information among many variables was first introduced as "total correlation" [3] and is also called multi-information [4] or multivariate mutual information [5].

$$TC(X_G) = \sum_{i \in G} H(X_i) - H(X_G) \tag{1}$$

For $G = \{i_1, i_2\}$, this corresponds to the mutual information, $I(X_{i_1} : X_{i_2})$. $TC(X_G)$ is non-negative and zero if and only if the probability distribution factorizes. In fact, total correlation can also be written as a KL divergence, $TC(X_G) = D_{KL}(p(x_G) || \prod_{i \in G} p(x_i))$.

The total correlation among a group of variables, $X$, after conditioning on some other variable, $Y$, is simply $TC(X|Y) = \sum_i H(X_i|Y) - H(X|Y)$. We can measure the extent to which $Y$ *explains* the correlations in $X$ by looking at how much the total correlation is reduced.

$$TC(X;Y) \equiv TC(X) - TC(X|Y) = \sum_{i \in \mathbb{N}_n} I(X_i : Y) - I(X : Y) \tag{2}$$

We use semicolons as a reminder that $TC(X;Y)$ is not symmetric in the arguments, unlike mutual information. $TC(X|Y)$ is zero (and $TC(X;Y)$ maximized) if and only if the distribution of $X$'s conditioned on $Y$ factorizes. This would be the case if $Y$ were the common cause of all the $X_i$'s in which case $Y$ *explains* all the correlation in $X$. $TC(X_G|Y) = 0$ can also be seen as encoding local Markov properties among a group of variables and, therefore, specifying a DAG [6]. This quantity has appeared as a measure of the *redundant* information that the $X_i$'s carry about $Y$ [7]. More connections are discussed in Sec. 5.

Optimizing over Eq. 2 can now be seen as a search for a latent factor, $Y$, that explains the correlations in $X$. We can make this concrete by letting $Y$ be a discrete random variable that can take one of $k$ possible values and searching over all probabilistic functions of $X$, $p(y|x)$.

$$\max_{p(y|x)} TC(X;Y) \quad \text{s.t.} \quad |Y| = k, \tag{3}$$

The solution to this optimization is given as a special case in Sec. A. Total correlation is a functional over the joint distribution, $p(x, y) = p(y|x)p(x)$, so the optimization implicitly depends on the data through $p(x)$. Typically, we have only a small number of samples drawn from $p(x)$ (compared to the size of the state space). To make matters worse, if $x \in \{0, 1\}^n$ then optimizing over all $p(y|x)$ involves at least $2^n$ variables. Surprisingly, despite these difficulties we show in the next section that this optimization can be carried out efficiently. The maximum achievable value of this objective occurs for some finite $k$ when $TC(X|Y) = 0$. This implies that the data are perfectly described by a naive Bayes model with $Y$ as the parent and $X_i$ as the children.

Generally, we expect that correlations in data may result from several different factors. Therefore, we extend the optimization above to include $m$ different factors, $Y_1, \ldots, Y_m$.[1]

$$\max_{G_j, p(y_j|x_{G_j})} \sum_{j=1}^{m} TC(X_{G_j}; Y_j) \quad \text{s.t.} \quad |Y_j| = k, G_j \cap G_{j' \neq j} = \emptyset \tag{4}$$

Here we simultaneously search subsets of variables $G_j$ and over variables $Y_j$ that explain the correlations in each group. While it is not necessary to make the optimization tractable, we impose an additional condition on $G_j$ so that each variable $X_i$ is in a single group, $G_j$, associated with a single "parent", $Y_j$. The reason for this restriction is that it has been shown that the value of the objective can then be interpreted as a lower bound on $TC(X)$ [8]. Note that this objective is valid

and meaningful regardless of details about the data-generating process. We only assume that we are given $p(x)$ or iid samples from it.

The output of this procedure gives us $Y_j$'s, which are probabilistic functions of $X$. If we iteratively apply this optimization to the resulting probability distribution over $Y$ by searching for some $Z_1, \ldots, Z_{\tilde{m}}$ that explain the correlations in the $Y$'s, we will end up with a hierarchy of variables that forms a tree. We now show that the optimization in Eq. 4 can be carried out efficiently even for high-dimensional spaces and small numbers of samples.

## 3  CorEx: Efficient Implementation of Correlation Explanation

We begin by re-writing the optimization in Eq. 4 in terms of mutual informations using Eq. 2.

$$\max_{G, p(y_j|x)} \sum_{j=1}^{m} \sum_{i \in G_j} I(Y_j : X_i) - \sum_{j=1}^{m} I(Y_j : X_{G_j}) \tag{5}$$

Next, we replace $G$ with a set indicator variable, $\alpha_{i,j} = \mathbb{I}[X_i \in G_j] \in \{0, 1\}$.

$$\max_{\alpha, p(y_j|x)} \sum_{j=1}^{m} \sum_{i=1}^{n} \alpha_{i,j} I(Y_j : X_i) - \sum_{j=1}^{m} I(Y_j : X) \tag{6}$$

The non-overlapping group constraint is enforced by demanding that $\sum_{\bar{j}} \alpha_{i,\bar{j}} = 1$. Note also that we dropped the subscript $G_j$ in the second term of Eq. 6 but this has no effect because solutions must satisfy $I(Y_j : X) = I(Y_j : X_{G_j})$, as we now show.

For fixed $\alpha$, it is straightforward to find the solution of the Lagrangian optimization problem as the solution to a set of self-consistent equations. Details of the derivation can be found in Sec. A.

$$p(y_j|x) = \frac{1}{Z_j(x)} p(y_j) \prod_{i=1}^{n} \left( \frac{p(y_j|x_i)}{p(y_j)} \right)^{\alpha_{i,j}} \tag{7}$$

$$p(y_j|x_i) = \sum_{\bar{x}} p(y_j|\bar{x}) p(\bar{x}) \delta_{\bar{x}_i, x_i} / p(x_i) \text{ and } p(y_j) = \sum_{\bar{x}} p(y_j|\bar{x}) p(\bar{x}) \tag{8}$$

Note that $\delta$ is the Kronecker delta and that $Y_j$ depends only on the $X_i$ for which $\alpha_{i,j}$ is non-zero. Remarkably, $Y_j$'s dependence on $X$ can be written in terms of a *linear* (in $n$, the number of variables) number of parameters which are just the marginals, $p(y_j), p(y_j|x_i)$. We approximate $p(x)$ with the empirical distribution, $\hat{p}(\bar{x}) = \sum_{l=1}^{N} \delta_{\bar{x}, x^{(l)}} / N$. This approximation allows us to estimate marginals with fixed accuracy using only a constant number of iid samples from the true distribution. In Sec. A we show that Eq. 7, which defines the soft labeling of any $x$, can be seen as a linear function followed by a non-linear threshold, reminiscent of neural networks. Also note that the normalization constant for any $x$, $Z_j(x)$, can be calculated easily by summing over just $|Y_j| = k$ values.

For fixed values of the parameters $p(y_j|x_i)$, we have an integer linear program for $\alpha$ made easy by the constraint $\sum_{\bar{j}} \alpha_{i,\bar{j}} = 1$. The solution is $\alpha_{i,j}^* = \mathbb{I}[j = \arg\max_{\bar{j}} I(X_i : Y_{\bar{j}})]$. However, this leads to a rough optimization space. The solution in Eq. 7 is valid (and meaningful, see Sec. 5 and [8]) for arbitrary values of $\alpha$ so we relax our optimization accordingly. At step $t = 0$ in the optimization, we pick $\alpha_{i,j}^{t=0} \sim \mathcal{U}(1/2, 1)$ uniformly at random (violating the constraints). At step $t + 1$, we make a small update on $\alpha$ in the direction of the solution.

$$\alpha_{i,j}^{t+1} = (1 - \lambda) \alpha_{i,j}^t + \lambda \alpha_{i,j}^{**} \tag{9}$$

The second term, $\alpha_{i,j}^{**} = \exp\left(\gamma(I(X_i : Y_j) - \max_{\bar{j}} I(X_i : Y_{\bar{j}}))\right)$, implements a soft-max which converges to the true solution for $\alpha^*$ in the limit $\gamma \to \infty$. This leads to a smooth optimization and good choices for $\lambda, \gamma$ can be set through intuitive arguments described in Sec. B.

Now that we have rules to update both $\alpha$ and $p(y_j|x_i)$ to increase the value of the objective, we simply iterate between them until we achieve convergence. While there is no guarantee to find the global optimum, the objective is upper bounded by $TC(X)$ (or equivalently, $TC(X|Y)$ is lower bounded by 0). Pseudo-code for this approach is described in Algorithm 1 with additional details provided in Sec. B and source code available online[2]. The overall complexity is linear in the number

**Algorithm 1:** Pseudo-code implementing Correlation Explanation (CorEx)

of variables. To bound the complexity in terms of the number of samples, we can always use mini-batches of fixed size to estimate the marginals in Eq. 8.

A common problem in representation learning is how to pick $m$, the number of latent variables to describe the data. Consider the limit in which we set $m = n$. To use all $Y_1, \ldots, Y_m$ in our representation, we would need exactly one variable, $X_i$, in each group, $G_j$. Then $\forall j, TC(X_{G_j}) = 0$ and, therefore, the whole objective will be 0. This suggests that the maximum value of the objective must be achieved for some value of $m < n$. In practice, this means that if we set $m$ too high, only some subset of latent variables will be used in the solution, as we will demonstrate in Fig. 2. In other words, if $m$ is set high enough, the optimization will result in some number of clusters $m' < m$ that is optimal with respect to the objective. Representations with different numbers of layers, different $m$, and different $k$ can be compared according to how tight of a lower bound they provide on $TC(X)$ [8].

## 4 Experiments

### 4.1 Synthetic data

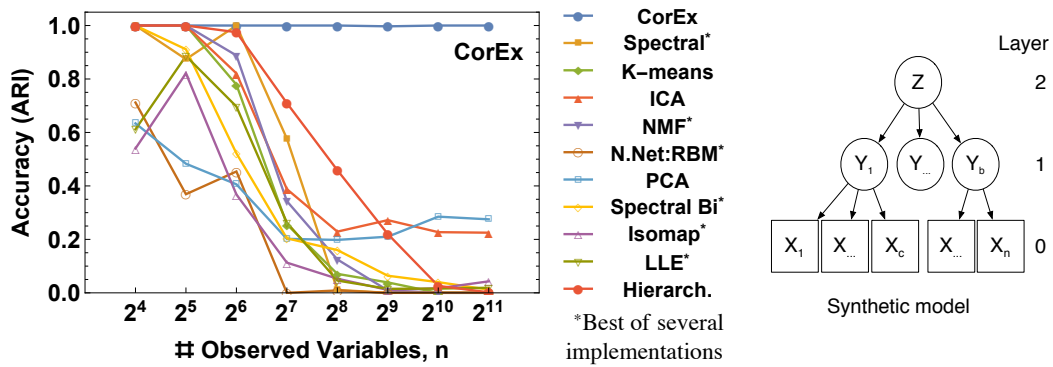

Figure 1: (Left) We compare methods to recover the clusters of variables generated according to the model. (Right) Synthetic data is generated according to a tree of latent variables.

To test CorEx's ability to recover latent structure from data we begin by generating synthetic data according to the latent tree model depicted in Fig. 1 in which all the variables are hidden except for the leaf nodes. The most difficult part of reconstructing this tree is clustering of the leaf nodes. If a clustering method can do that then the latent variables can be reconstructed for each cluster easily using EM. We consider many different clustering methods, typically with several variations

of each technique, details of which are described in Sec. C. We use the adjusted Rand index (ARI) to measure the accuracy with which inferred clusters recover the ground truth. [3]

We generated samples from the model in Fig. 1 with $b = 8$ and varied $c$, the number of leaves per branch. The $X_i$'s depend on $Y_j$'s through a binary erasure channel (BEC) with erasure probability $\delta$. The capacity of the BEC is $1 - \delta$ so we let $\delta = 1 - 2/c$ to reflect the intuition that the signal from each parent node is weakly distributed across all its children (but cannot be inferred from a single child). We generated $\max(200, 2n)$ samples. In this example, all the $Y_j$'s are weakly correlated with the root node, $Z$, through a binary symmetric channel with flip probability of $1/3$.

Fig. 1 shows that for a small to medium number of variables, all the techniques recover the structure fairly well, but as the dimensionality increases only CorEx continues to do so. ICA and hierarchical clustering compete for second place. CorEx also perfectly recovers the values of the latent factors in this example. For latent tree models, recovery of the latent factors gives a global optimum of the objective in Eq. 4. Even though CorEx is only guaranteed to find local optima, in this example it correctly converges to the global optimum over a range of problem sizes.

Note that a growing literature on latent tree learning attempts to reconstruct latent trees with theoretical guarantees [9, 1]. In principle, we should compare to these techniques, but they scale as $O(n^2) - O(n^5)$ (see [3], Table 1) while our method is $O(n)$. In a recent survey on latent tree learning methods, only one out of 15 techniques was able to run on the largest dataset considered (see [3], Table 3), while most of the datasets in this paper are orders of magnitude larger than that one.

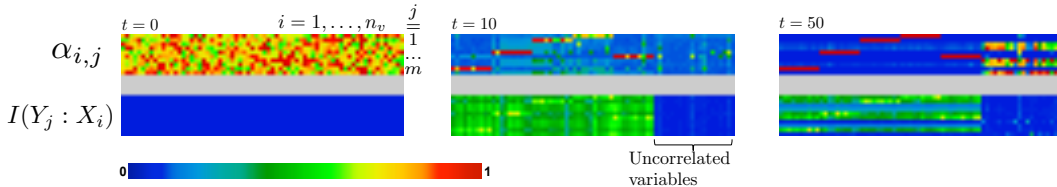

Figure 2: (Color online) A visualization of structure learning in CorEx, see text for details.

Fig. 2 visualizes the structure learning process. [4] This example is similar to that above but includes some uncorrelated random variables to show how they are treated by CorEx. We set $b = 5$ clusters of variables but we used $m = 10$ hidden variables. At each iteration, $t$, we show which hidden variables, $Y_j$, are connected to input variables, $X_i$, through the connectivity matrix, $\alpha$ (shown on top). The mutual information is shown on the bottom. At the beginning, we started with full connectivity, but with nothing learned we have $I(Y_j : X_i) = 0$. Over time, the hidden units "compete" to find a group of $X_i$'s for which they can explain all the correlations. After only ten iterations the overall structure appears and by 50 iterations it is exactly described. At the end, the uncorrelated random variables ($X_i$'s) and the hidden variables ($Y_j$'s) which have not explained any correlations can be easily distinguished and discarded (visually and mathematically, see Sec. B).

## 4.2   Discovering Structure in Diverse Real-World Datasets

### 4.2.1   Personality Surveys and the "Big Five" Personality Traits

One psychological theory suggests that there are five traits that largely reflect the differences in personality types [1]: *extraversion*, *neuroticism*, *agreeableness*, *conscientiousness* and *openness to experience*. Psychologists have designed various instruments intended to measure whether individuals exhibit these traits. We consider a survey in which subjects rate fifty statements, such as, "I am the life of the party", on a five point scale: (1) disagree, (2) slightly disagree, (3) neutral, (4) slightly agree, and (5) agree. [5] The data consist of answers to these questions from about ten thousand test-takers. The test was designed with the intention that each question should belong to a

cluster according to which personality trait the question gauges. Is it true that there are five factors that strongly predict the answers to these questions?

CorEx learned a two-level hierarchical representation when applied to this data (full model shown in Fig. C.2). On the first level, CorEx automatically determined that the questions should cluster into five groups. Surprisingly, the five clusters *exactly* correspond to the big five personality traits as labeled by the test designers. It is unusual to recover the ground truth with perfect accuracy on an unsupervised learning problem so we tried a number of other standard clustering methods to see if they could reproduce this result. We display the results using confusion matrices in Fig. 3. The details of the techniques used are described in Sec. C but all of them had an advantage over CorEx since they required that we specify the correct number of clusters. None of the other techniques are able to recover the five personality types exactly.

Interestingly, Independent Component Analysis (ICA) [1] is the only other method that comes close. The intuition behind ICA is that it find a linear transformation on the input that minimizes the multi-information among the outputs ($Y_j$). In contrast, CorEx searches for $Y_j$'s so that multi-information among the $X_i$'s is minimized after conditioning on $Y$. ICA assumes that the signals that give rise to the data are independent while CorEx does not. In this case, personality traits like "extraversion" and "agreeableness" are correlated, violating the independence assumption.

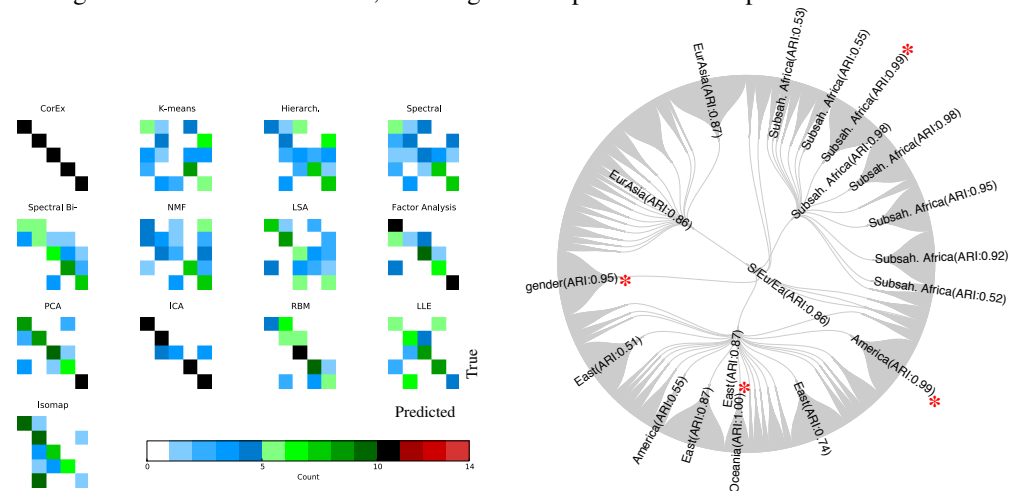

Figure 3: (Left) Confusion matrix comparing predicted clusters to true clusters for the questions on the Big-5 personality test. (Right) Hierarchical model constructed from samples of DNA by CorEx.

### 4.2.2 DNA from the Human Genome Diversity Project

Next, we consider DNA data taken from 952 individuals of diverse geographic and ethnic backgrounds [1]. The data consist of 4170 variables describing different SNPs (single nucleotide polymorphisms).[6] We use CorEx to learn a hierarchical representation which is depicted in Fig. 3. To evaluate the quality of the representation, we use the adjusted Rand index (ARI) to compare clusters induced by each latent variable in the hierarchical representation to different demographic variables in the data. Latent variables which substantially match demographic variables are labeled in Fig. 3.

The representation learned (*unsupervised*) on the first layer contains a perfect match for Oceania (the Pacific Islands) and nearly perfect matches for America (Native Americans), Subsaharan Africa, and gender. The second layer has three variables which correspond very closely to broad geographic regions: Subsaharan Africa, the "East" (including China, Japan, Oceania, America), and EurAsia.

### 4.2.3 Text from the Twenty Newsgroups Dataset

The twenty newsgroups dataset consists of documents taken from twenty different topical message boards with about a thousand posts each [1]. For analyzing unstructured text, typical feature engineering approaches heuristically separate signals like style, sentiment, or topics. In principle, all

three of these signals manifest themselves in terms of subtle correlations in word usage. Recent attempts at learning large-scale unsupervised hierarchical representations of text have produced interesting results [1], though validation is difficult because quantitative measures of representation quality often do not correlate well with human judgment [1].

To focus on linguistic signals, we removed meta-data like headers, footers, and replies even though these give strong signals for supervised newsgroup classification. We considered the top ten thousand most frequent tokens and constructed a bag of words representation. Then we used CorEx to learn a five level representation of the data with 326 latent variables in the first layer. Details are described in Sec. C.1. Portions of the first three levels of the tree keeping only nodes with the highest normalized mutual information with their parents are shown in Fig. 4 and in Fig. C.1.[7]

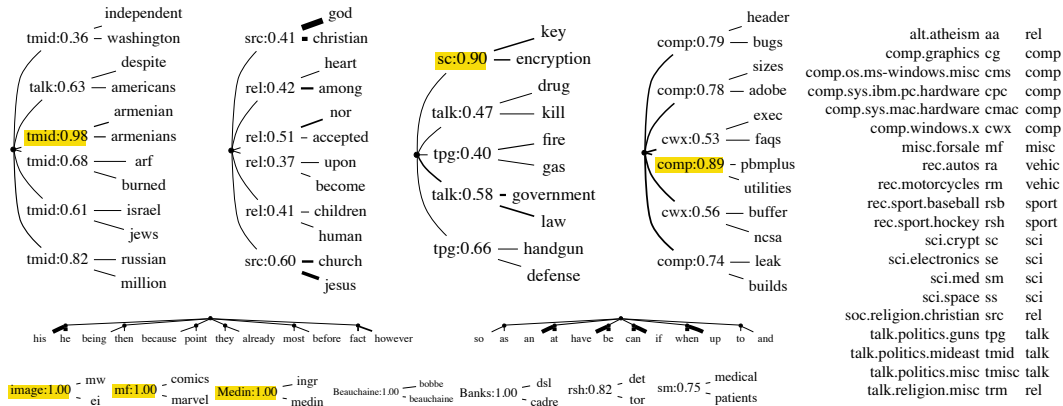

Figure 4: Portions of the hierarchical representation learned for the twenty newsgroups dataset. We label latent variables that overlap significantly with known structure. Newsgroup names, abbreviations, and broad groupings are shown on the right.

To provide a more quantitative benchmark of the results, we again test to what extent learned representations are related to known structure in the data. Each post can be labeled by the newsgroup it belongs to, according to broad categories (e.g. groups that include "comp"), or by author. Most learned binary variables were active in around 1% of the posts, so we report the fraction of activations that coincide with a known label (precision) in Fig. 4. Most variables clearly represent sub-topics of the newsgroup topics, so we do not expect high recall. The small portion of the tree shown in Fig. 4 reflects intuitive relationships that contain hierarchies of related sub-topics as well as clusters of function words (e.g. pronouns like "he/his/him" or tense with "have/be").

Once again, several learned variables perfectly captured known structure in the data. Some users sent images in text using an encoded format. One feature matched all the image posts (with perfect precision and recall) due to the correlated presence of unusual short tokens. There were also perfect matches for three frequent authors: G. Banks, D. Medin, and B. Beauchaine. Note that the learned variables did not trigger if just their names appeared in the text, but only for posts they authored. These authors had elaborate signatures with long, identifiable quotes that evaded preprocessing but created a strongly correlated signal. Another variable with perfect precision for the "forsale" newsgroup labeled comic book sales (but did not activate for discussion of comics in other newsgroups). Other nearly perfect predictors described extensive discussions of Armenia/Turkey in *talk.politics.mideast* (a fifth of all discussion in that group), specialized unix jargon, and a match for *sci.crypt* which had 90% precision and 55% recall. When we ranked all the latent factors according to a normalized version of Eq. 2, these examples all showed up in the top 20.

## 5 Connections and Related Work

While the basic measures used in Eq. 1 and Eq. 2 have appeared in several contexts [7, 1, 4, 3, 1], the interpretation of these quantities is an active area of research [1, 2]. The optimizations we define have some interesting but less obvious connections. For instance, the optimization in Eq. 3 is similar

to one recently introduced as a measure of "common information" [2]. The objective in Eq. 6 (for a single $Y_j$) appears exactly as a bound on "ancestral" information [2]. For instance, if all the $\alpha_i = 1/\beta$ then Steudel and Ay [2] show that the objective is positive only if at least $1 + \beta$ variables share a common ancestor in any DAG describing them. This provides extra rationale for relaxing our original optimization to include non-binary values of $\alpha_{i,j}$.

The most similar learning approach to the one presented here is the information bottleneck [2] and its extension the multivariate information bottleneck [2, 2]. The motivation behind information bottleneck is to compress the data ($X$) into a smaller representation ($Y$) so that information about some relevance term (typically labels in a supervised learning setting) is maintained. The second term in Eq. 6 is analogous to the compression term. Instead of maximizing a relevance term, we are maximizing information about all the individual sub-systems of $X$, the $X_i$. The most redundant information in the data is preferentially stored while uncorrelated random variables are completely ignored.

The broad problem of transforming complex data into simpler, more meaningful forms goes under the rubric of *representation learning* [2] which shares many goals with dimensionality reduction and subspace clustering. Insofar as our approach learns a hierarchy of representations it superficially resembles "deep" approaches like neural nets and autoencoders [2, 2, 2, 3]. While those approaches are scalable, a common critique is that they involve many heuristics discovered through trial-and-error that are difficult to justify. On the other hand, a rich literature on learning latent tree models [3, 3, 9, 1] have excellent theoretical properties but do not scale well. By basing our method on an information-theoretic optimization that can nevertheless be performed quite efficiently, we hope to preserve the best of both worlds.

## 6   Conclusion

The most challenging open problems today involve high-dimensional data from diverse sources including human behavior, language, and biology.[8] The complexity of the underlying systems makes modeling difficult. We have demonstrated a model-free approach to learn successfully more coarse-grained representations of complex data by efficiently optimizing an information-theoretic objective. The principle of explaining as much correlation in the data as possible provides an intuitive and fully data-driven way to discover previously inaccessible structure in high-dimensional systems.

It may seem surprising that CorEx should perfectly recover structure in diverse domains without using labeled data or prior knowledge. On the other hand, the patterns discovered are "low-hanging fruit" from the right point of view. Intelligent systems should be able to learn robust and general patterns in the face of rich inputs even in the absence of labels to define what is important. Information that is very redundant in high-dimensional data provides a good starting point.

Several fruitful directions stand out. First, the promising preliminary results invite in-depth investigations on these and related problems. From a computational point of view, the main work of the algorithm involves a matrix multiplication followed by an element-wise non-linear transform. The same is true for neural networks and they have been scaled to very large data using, e.g., GPUs. On the theoretical side, generalizing this approach to allow non-tree representations appears both feasible and desirable [8].

**Acknowledgments**

We thank Virgil Griffith, Shuyang Gao, Hsuan-Yi Chu, Shirley Pepke, Bilal Shaw, Jose-Luis Ambite, and Nathan Hodas for helpful conversations. This research was supported in part by AFOSR grant FA9550-12-1-0417 and DARPA grant W911NF-12-1-0034.

## Footnotes

[1]Note that in principle we could have just replaced $Y$ in Eq. 3 with $(Y_1, \ldots, Y_m)$, but the state space would have been exponential in $m$, leading to an intractable optimization.

[2] Open source code is available at `http://github.com/gregversteeg/CorEx`.

[3]Rand index counts the percentage of pairs whose relative classification matches in both clusterings. ARI adds a correction so that a random clustering will give a score of zero, while an ARI of 1 corresponds to a perfect match.

[4]A video is available online at `http://isi.edu/~gregv/corex_structure.mpg`.

[5]Data and full list of questions are available at `http://personality-testing.info/_rawdata/`.

[6]Data, descriptions of SNPs, and detailed demographics of subjects is available at `ftp://ftp.cephb.fr/hgdp_v3/`.

[7]An interactive tool for exploring the full hierarchy is available at `http://bit.ly/corexvis`.

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
