[Supplementary Material · corex_app_supp.pdf]

## Supplementary Material for "Discovering Structure in High-Dimensional Data Through Correlation Explanation"

## A  Derivation of Eqs. 7 and 8

We want to optimize the following objective.

$$\max_{\alpha, p(y_j|x)} \sum_{j=1}^{m} \sum_{i=1}^{n} \alpha_{i,j} I(Y_j : X_i) - \sum_{j=1}^{m} I(Y_j : X)$$

$$\text{s.t.} \sum_{y_j} p(y_j|x) = 1 \tag{10}$$

In principle, we would also like $\forall i, j, \alpha_{i,j} \in \{0,1\}, \sum_{\bar{j}} \alpha_{i,\bar{j}} = 1$, but we begin by solving the optimization for fixed $\alpha$.

We proceed using Lagrangian optimization. We introduce a Lagrange multiplier $\lambda_j(x)$ for each value of $x$ and each $j$ to enforce the normalization constraint and then reduce the constrained optimization problem to the unconstrained optimization of the objective $\mathcal{L}_{tot} = \sum_j \mathcal{L}_j$. We show the solution for a single $\mathcal{L}_j$, but drop the $j$ index to avoid clutter. (For fixed $\alpha$, the optimization for different $j$ totally decouple.)

$$\mathcal{L} = \sum_{x,y} p(x)p(y|x) \left( \sum_i \alpha_i (\log p(y|x_i) - \log(p(y))) - (\log p(y|x) - \log(p(y))) \right)$$

$$+ \sum_x \lambda(x) (\sum_y p(y|x) - 1)$$

Note that we are optimizing over $p(y|x)$ and so the marginals $p(y|x_i), p(y)$ are actually linear functions of $p(y|x)$. Next we take the functional derivatives with respect to $p(y|x)$ and set them equal to 0. Note that this can be done symbolically and proceeds in similar fashion to the detailed calculations of information bottleneck [25].

This leads to the following condition.

$$p(y_j|x) = \frac{1}{Z(x)} p(y_j) \prod_{i=1}^{n} \left( \frac{p(y_j|x_i)}{p(y_j)} \right)^{\alpha_{i,j}}$$

But this is only a formal solution since the marginals themselves are defined in terms of $p(y|x)$.

$$p(y) = \sum_x p(x)p(y|x), \quad p(y|x_i) = \sum_{x_{j \neq i}} p(y|x)p(x)/p(x_i)$$

The partition constant, $Z(x)$ can be easily calculated by summing over just $|Y_j|$ terms.

Imagine we are given $l = 1, \ldots, N$ samples, $x^{(l)}$, drawn from unknown distribution $p(x)$. If $x$ is very high dimensional, we do not want to enumerate over all possible values of $x$. Instead, we consider the quantity in Eq. 7 and Eq. 8 only for observed samples.

$$p(y|x^{(l)}) = \frac{1}{Z(x^{(l)})} p(y) \prod_{i=1}^{n} \left( \frac{p(y|x_i^{(l)})}{p(y)} \right)^{\alpha_{i,j}}$$

In log-space, this has an even simpler form.

$$\log p(y|x^{(l)}) = (1 - \sum_i \alpha_i) \log p(y) + \sum_{i=1}^{n} \alpha_i \log p(y|x_i^{(l)}) - \log Z(x^{(l)})$$

That is, the probabilistic label, $y$, for any sample, $x$, is a linear combination of weighted terms for each $x_i$. We recover $p(y|x)$ by doing a nonlinear transformation consisting of exponentiation and normalization.

The consistency requirements which are sums over the state space of $x$ can be replaced with sample expectations.

$$p(y) = \sum_x p(x)p(y|x) \approx \frac{1}{N}\sum_{j=1}^N p(y|x^{(l)}),$$

with similar estimates for the marginals $p(y|x_i)$. In practice, to limit the complexity in terms of the number of samples, we can choose a random subset of samples at each iteration and estimate the probabilistic labels and marginals only for them. The details of the optimization over $\alpha$ are described in the next section.

**Special case for Eq. 3** Note that the optimization in Eq. 3 corresponds to $j = 1, \ldots, m$ with $m = 1$ and $\forall i, \alpha_i = 1$.

**Convergence** The updates for the iterative procedure described here are guaranteed not to decrease the objective at each step and are guaranteed to converge to a local optimum. Theoretical details are described elsewhere [8].

## B  Implementation Details for CorEx

As pointed out in Sec. 5, the objective in Eq. 6 (for a single $Y_j$) appears exactly as a bound on "ancestral" information [22]. We use this fact to motivate our choice for parameters in Eq. 9. Consider the soft-max function we use to define $\alpha^*$.

$$\alpha^*_{i,j} = \exp\left(\gamma(I(X_i : Y_j) - \max_{\bar{j}} I(X_i : Y_{\bar{j}}))\right)$$

First of all, we allow $\gamma_{i,j}$ to take different values at different $i, j$. We start by enforcing the form $\gamma_{i,j} = C_j/H(X_i)$. That way, the value of the exponent depends on normalized mutual information (NMI) instead of mutual information. The minimum value that can occur is $\exp(-C_j)$. We set $C_j = 1$. If the difference of $NMI$'s take the minimum value of $-1$, we get $\alpha^*_{i,j} \sim 1/3$. According to the Steudel and Ay bound, $X_i$ can still contribute to a non-negative value for the part of objective Eq. 6 that involves $Y_j$ as long as $X_i$ shares a common ancestor with at least $1/\alpha + 1$ other variables. At the beginning of the learning, this is desirable as it allows all $Y_j$'s to learn significant structures even starting from small values of $\alpha_{i,j}$. However, as the computation progresses, we would like to force the soft-max function to get closer to the true hard max solution. To that end, we set $\gamma_{i,j} = (1 + D_j)/H(X_i)$, where $D_j = 500 \cdot |\mathbb{E}_X(-\log Z_j(x))|$. The $D_j$ term represents the amount of correlation learned by $Y_j$ [8]. For instance, if all $p(y_j|x_i) = p(y_j)$, $\log Z_j(x) = 0$ and $Y_j$ has not learned anything. As the computation progresses and $Y_j$ learns more structure, we smoothly transition to a hard-max constraint.

In all the experiments shown here, we set $|Y_j| = k = 2$. For convergence of Algorithm 1, we check when the magnitude of changes of $\mathbb{E}_X \log(-\sum_j Z_j(x))$ consistently falls below a threshold of $10^{-5}$ or when we reach 1000 iterations, whichever occurs first. We set $\lambda = 0.3$ based on several tests with synthetic data.

We construct higher order representation from the bottom up. After applying Algorithm 1, we take the most likely value of $Y_j$ for each sample in the dataset. Then we apply CorEx again using these labels as the input. In principle, this sample of $Y$'s does not accurately reflect $p(y) = \sum_x p(y|x)p(x)$ and a more nuanced approximation like contrastive divergence could be used. However, in practice it seems that CorEx typically learns nearly deterministic functions of $x$, so that the maximum likelihood labels well reflect the true distribution.

In Fig. 2, we suggested that uncorrelated random variables could be easily detected. In practice we used a threshold that this was the case if $MI(X_i : Y_{parent(X_i)})/min(H(X_i), H(Y)) < 0.05$. At higher layers of representation, this helps us identify root nodes. For the DNA example in Fig. 3, "gender" was a root node, but for visual simplicity all root nodes were connected at the top level. Following similar reasoning as above, we can also check which $Y_j$'s have learned significant structure by looking at the value of $\mathbb{E}_X(-\log Z_j(x))$.

# C   Implementation Details for Comparisons

We represented the data from the binary erasure channel either as integers $(0[X_i = 0], 1[X_i = e], 2[X_i = 1])$ for methods that deal with categorical data, or as floating numbers on the unit interval for methods that require data of that form, $(0[X_i = 0], 0.5[X_i = e], 1[X_i = 1])$. In principle, we could also have treated "erased" information as missing. But we treated erasure as another outcome in all cases, including for CorEx.

CorEx naturally handles missing information (you can see that Eq. 7 can be easily marginalized to find labels even if some variables are missing). We had to use this fact for the DNA dataset which did have some SNPs missing for some samples. In fact, because CorEx is, in a sense, looking for the most redundant information, it is quite robust to missing information.

We will now briefly describe the settings for various learning algorithms learned. We used implementations of standard learning techniques in the scikit library for comparisons [34] (v. 0.14). We only used the standard, default implementation for k-means, PCA, ICA, and "hierarchical clustering" using the Ward method. For spectral clustering we used a Gaussian kernel for the affinity matrix and a nearest neighbors affinity matrix using 3 or 10 neighbors. For spectral bi-clustering we tried clustering either the data matrix or its transpose. We set the number of clusters to be $m$ in the direction of variables and either 10 or 32 clusters for the variables. Note that the true number of clusterings in the sample space was $2^8$. For NMF we tried Projected Gradient NMF and NMF with the two types of implemented sparseness constraints. For the restricted Boltzmann machine, we used a single layer network with $m$ units and learning rates $0.01, 0.05, 0.1$. To cluster the input variables, we looked for the neuron with the maximum magnitude weight. For dimensionality reduction techniques like LLE and Isomap, we used either 3 or 10 nearest neighbors and looked for a $m$ component representation. Then we clustered variables by looking at which variables contributed most to each component of the representation.

## C.1   Twenty newsgroups

For the twenty newsgroups dataset, scikit has built-in function for retrieving and processing the dataset. We used the command below, resulting in a dataset with $18,846$ posts. (Several different versions of this dataset are in circulation.)

```
sklearn.datasets.fetch_20newsgroups(subset='all',
            remove=('headers','footers','quotes')).
```

Because we are doing unsupervised learning, we combined the parts of the data normally split into training and testing sets. The attempt to strip footers turned out to be particularly relevant. The heuristic to do so looks for a single line at the end of the file, set apart from the others by a blank line or some number of dashes. Obviously, many signature lines fail to conform to this format and this resulted in strongly correlated signals. This led to features at layer 1 that were perfect predictors of authors, like Gordon Banks, who always included a quote: "Skepticism is the chastity of the intellect, and it is shameful to surrender it too soon."

We considered any collection of upper or lower-case letters as a "word". All characters were lower-cased. Apostrophes were removed (so that "I've" becomes "ive"). We considered the top ten thousand most frequent words. For the thousand most frequent words, for each document we recorded a 0 if the word was not present, 1 if it was present but occurred with less than the average frequency, or a 2 if it occurred with more than average frequency. For the remaining words we just used a 0/1 representation to reflect if a word was present.

**CorEx details**   For the twenty newsgroups data, we trained CorEx in a top-down-bottom-up way. We started with a "low resolution" model with $m = 100$ hidden units and $k = 2$. We used the result of this optimization to construct 100 large groups of words. Then, for each (now much smaller) group of words, we applied CorEx again to get a more fine-grained representation (and then we discard the representation that we used to find the original clustering). The result was a representation at layer 1 with 326 variables. At the next layer we fixed $m = 50$, all units were used. At the next two layers we fixed $m = 10, 1$, respectively.

Figure C.1: The bottom three layers of the hierarchical representation learned for the twenty newsgroups dataset, keeping only the three leaf nodes with the highest normalized mutual information with their parents and up to eight branches per node at layer 2. For latent variables, we list an abbreviation of the newsgroup it best corresponds to along with the precision. For a zoomable version online go to http://bit.ly/corexvis.

Figure C.2: CorEx learns a hierarchical representation from personality surveys with 50 questions. The number of latent nodes in the tree and number of levels are automatically determined. The question groupings at the first level exactly correspond to the "big five" personality traits. The prefix of each question indicates the trait test designers intended it to measure. The thickness of each edge represents mutual information between features and the size of each node represents the total correlation that the node captures about its children.