[Reviews · NeurIPS 2014]

Submitted by Assigned_Reviewer_30

This paper presents an unsupervised dimensionality reduction algorithm which is based on the information bottleneck (IB) method. The method optimizes a constrained objective which, like the IB method, is comprised of the mutual information criteria. The criteria are between a joint density of discrete observed variables and the densities of a set of discrete latent factors, and between factored densities of the observed variables and the latent factors' densities. The goal of the the objective function is to infer latent factors such that by conditioning on them, the observed variables can be factorized into subsets of minimally correlated elements. An iterative algorithm is derived for optimizing the objective which alternates between updating conditional densities over the latent factors given a set of observed training data (using fixed-point equations) and updating the assignments of the observed variables to one of the latent factors. In the numerical experiments the model is applied to learn latent tree structures on a number of benchmarks. The hierarchical structures are learned by stacking multiple instances of the model such that maximum a posteriori latent interpretations of a trained model become the observed variables for the next model instance on top.

The paper presents a simple method for unsupervised dimensionality reduction which scales linearly w.r.t. the number and cardinality of variables involved. The objective function it optimizes allows for an automatic selection of the number of latent factors to be employed. Experimental results on synthetic data as well as on benchmarks from different domains show that the stacked implementation of the method performs a decent job of learning plausible/close to ground-truth latent hierarchies as compared to the other methods. On the other hand, the work lacks details about the consistency of the proposed method. The model uses fixed-point updates for estimating posteriors over the latents which are set randomly at the beginning. Moreover, the values of alpha_{i,j} are also randomly initialized (violating constraints on them) and are then gradually updated using lambda and adaptive gamma_{i,j} parameters. To that end however, no analysis is performed on model's convergence behaviour e.g., for increasingly large number and cardinalities of model variables. Also details are lacking on the sampling of training data and it is not clear how much a chosen sampling regime may effect the quality of probability estimates and in turn overall parameter evolution and performance.

The method is proposed for discrete random variables and it also imposes the condition that each of the observed variables belongs to one latent factor, hence claiming that the model requires no assumptions is not actually true.

For the relevant work, how does the proposed method can be contrasted with the recent work of Song et al. ICML 2013, on hierarchical decomposition of latent tree structures?

Altogether the paper is well written and it is clear about conveying the main message and results.
Summary: The paper makes an incremental but sound theoretical contribution. Practical significance of the method is demonstrated on a variety of datasets, although some basic analysis about its training procedure remains lacking.

Submitted by Assigned_Reviewer_41

Summary of paper:

The paper presents an unsupervised technique for learning a hierarchy of latent factors. At each level of the hierarchy, they find a partially factorized mean-field approximation of the variables in the previous layer, with each variable ending up in one group. The final output is a tree-shaped model of the data.

At each level of the tree, they provide an efficient algorithm to optimize the objective by alternately optimizing the latent factor marginals and the group assignments.

In experiments, they demonstrate success in recovering a variety of latent structures from DNA and text, such as geographic information and topics, as well as reconstructing synthetic data.

Quality:

Paper is technically sound in their definition of the objective and optimization method. Intuition towards sample complexity is given but no formal results, which is fine. Many different clustering methods are compared to in their synthetic experiments, though comparisons against other methods on real data would be even better. Additionally, more detailed Related Work re: previous structure learning methods would be appreciated.

Clarity:

This is a well written paper, that spends a lot of time in each section attempting to give an intuitive feel for what the algorithm is doing. The inclusion of implementation details and derivations in the supplemental material is appreciated.

Originality:

The paper does not appear very different in spirit from other hierarchical clustering methods, but uses a novel objective function that seems to get better results than other clustering algorithms when the number of variables is large. The authors give a nice intuition about why this is the case - their objective function discards independent "noisy" variables.

Significance:

There is a lot of interest in learning more "principled" optimization methods for deep architectures. Despite the comparison to deep neural models, those often have non-tree-shaped convolutional architectures, whereas this paper optimizes tree-shaped architectures only. This appears to be more like a nice hierarchical clustering algorithm. The use of a multivariate information-bottleneck-like objective function for structure learning is a nice idea.
Summary: This is a well-executed but not particularly ground-breaking paper that presents a novel hierarchical clustering / factor analysis method based on an information bottleneck-type objective. The algorithm is technically sound, seems efficient, and the authors provide implementation details, several experiments, and good intuition as to the workings of the algorithm.

Submitted by Assigned_Reviewer_43

This paper is on finding common causes of dependencies in discrete multidimensional data using (conditional) total correlation, in order to provide a more compressed representation of the data.

The objective is to find the distribution of the "simplest", discrete, finite-valued random variable Y conditioned on which the KL divergence between the joint distribution over the data and the product of its marginals is minimized, where if I understand correctly, simplicity of Y is measured in terms of the different number k of values it can take. (It is not obvious how k is determined.) This is extended to include multiple common factors Y_1,...,Y_m which could explain the data. However, it is not obvious how the total number m of factors is obtained: a discussion is given on line 160, which I personally don't find convincing. In fact, while the idea of the paper is interesting, the main objective and the mathematical concepts need to be rigorously and formally stated. Such statements as "if we set m high enough, CorEx automatically chooses the optimal number of hidden variables to use (line 180)" need to be proven mathematically and under appropriate assumptions. In this particular case for example, finding m is analogous to finding the correct number of clusters in a clustering problem, which in certain frameworks is actually provably impossible.

The results are complemented with some interesting experiments on both synthetic and real data. However, I think that the comparisons with other methods may not be totally fair. For example, if I'm not mistaken, the Euclidean distance is used in k-means. However, since the objective is to find clusters of statistically dependent data, I personally think that if a measure of distance between probability distributions were used instead (e.g. the Wasserstein distance), the performance of k-means would be better. The results presented in 4.2.2 and 4.2.3 are interesting, however, it would be nice if they had been compared against other methods: This would give a general idea of how difficult it is to discover structures in the particular datasets used.

- minor comments -
* line 68: I think this should be changed to either TC(X_G) = [what's already written], or x_G must be replaced with x and G with N_n, keeping TC(X) as is.
* line 82: Is my interpretation correct? simplicity is indeed measured in terms of the value of k?
* typo on line 103: interpreted -> be interpreted
* in my opinion using \N_n for {1,...,n} can be confusing ...
* to the best of my knowledge, standard notation for mutual information is I(.;.) and not I(.:.).
* line 413: What do you mean by relaxing the tree structure?

Summary: The paper is interesting, however in my opinion it is not mature enough to be published at this point.
Author Feedback
Author rebuttal: We exhort all the reviewers to re-consider both the originality of this approach and the significance of the results on the "applications" axis of the NIPS review guidelines.

—Significance of Applications—

While the reviewers largely focused on (interesting) theoretical questions, we believe they greatly underestimated the significance and difficulty of the applications. For instance, while we included comparisons on synthetic and real-world data when possible, Reviewer_43 and 41 asked for comparisons on our larger real-world datasets. In a recent survey of latent tree-learning techniques, only one out of the 15 methods tested was able to run on a problem with 100 samples and 10k variables (Table 3, Mourad et al, 2013). The largest dataset we used was more than 100 times bigger (18k samples by 10k variables). Basically, no competitive methods exist for this task (at least none the authors know of nor that the reviewers pointed out).
The Song et.al. paper mentioned by Reviewer_30 reconstructs synthetic trees for up to 64 variables. We (perfectly) reconstructed synthetic trees many orders of magnitude larger than that. Our ability to do so is a significant achievement, even if we do not prove consistency results. The reason for this achievement comes from the scaling: previous latent tree techniques are at least O(n^2), ours is O(n).

Reviewer_43 refers only to "clustering", Reviewer_41 talks about "hierarchical clustering", and Reviewer_31 speaks of "dimensionality reduction". We want to emphasize that learning latent tree representations is a more general and difficult problem that goes beyond any of these subtasks.

Finally, human behavior, biology, and language represent three of the most difficult applications in machine learning. To get rich, useful results in all three domains without hyper-parameter tuning, supervision, or significant pre-processing is remarkable. For comparison, successful neural network applications are typically highly tuned to give good performance in a single domain, e.g., object recognition.

—Originality—

Previous latent tree learning methods require calculation over pairs, triples, or quartets of observed variables, leading to at least O(n^2) complexity (Table 1, Mourad et al, 2013). At best, those methods can be incrementally improved through clever pruning or parallelization (e.g., arXiv:1406.4566). Our approach achieves O(n) scalability by considering a radically different approach that does not use distances between observed variables at all.

More generally, most unsupervised learning attempts to maximize the likelihood of the data under some model. In our approach, at no point is a generative model nor any structure of the data-generating process assumed. The information-theoretic objective in Eq. 4 has a concrete meaning regardless of the generative model of the data (i.e. whether it is a tree or not).
The real question (not raised by the reviewers) is whether the value of this optimum lines up with a useful quantity for a particular model. While we do not attempt to answer this question for the menagerie of models where it might be of interest, the outstanding empirical results in very disparate systems (synthetic trees, human behavior, biology, language) easily justify reporting and further study.

— Other points —
We thank the reviewers for pointing out several small issues which we will clarify.
- Reviewer_43 argued that we did not prove "optimality" of the number of clusters. We should clarify that we are speaking about the "optimal" number of clusters w.r.t. our objective function. For many clustering methods, increasing the number of clusters always increases the score of the clustering objective (so that an extra penalty term is needed to pick the number of clusters). What is notable here is that this is not the case for our method. While the "optimal" number of clusters w.r.t. our objective need not line up with "optimality" for a particular clustering task (as Rev_43 points out, this may be impossible in some cases), empirically we show that the number of clusters found is correct for synthetic trees and lines up with ground truth on a real-world problem.
- To clarify Reviewer_30's question about assumptions: of course, we assume iid samples of discrete r.v.'s. We will clarify that "no assumptions" are made about the structure or form of the data-generating model, as discussed above. This is in contrast to standard tree learning techniques that assume the data is generated according to a tree to begin with.
- Reviewer_30 asks for convergence analysis. For synthetic tree models, the global optimum of Eq. 4 is achieved when Y_estimate = Y_tree. Therefore, Fig. 1 demonstrates that the global optimum is indeed obtained over several orders of magnitude of problem size.

This work makes scalable, principled learning of latent tree representations possible for the first time. Our very recent results on gene expression suggest that this will have significant ramifications for rich data exploration in some domains.